# Relationships between Foliar Fungal Endophyte Communities and Ecophysiological Traits of CAM and C$_3$ Epiphytic Bromeliads in a Neotropical Rainforest

**Peter H. Tellez [1],\*, Carrie L. Woods [2] , Stephen Formel [1] and Sunshine A. Van Bael [1,3]**

[1]  Department of Ecology and Evolutionary Biology, Tulane University, New Orleans, LA 70118, USA; sformel@tulane.edu (S.F.); svanbael@tulane.edu (S.A.V.B.)
[2]  Department of Biology, University of Puget Sound, Tacoma, WA 98416, USA; cwoods@pugetsound.edu
[3]  Smithsonian Tropical Research Institute, Balboa 0843-03092, Panama
\*  Correspondence: ptellez@tulane.edu; Tel.: +1-504-410-6939

**Abstract:** Vascular epiphytes contribute up to 35% of the plant diversity and foliar biomass of flowering plants. The family Bromeliaceae is a monophyletic group of plants native to the Neotropics. Epiphytic bromeliads form associations with distinct groups of organisms but their relationship with foliar fungal endophytes remain underexplored. In this study we examined the relationship of foliar fungal endophytes to host photosynthetic pathways and associated ecophysiological traits. We sampled the fungal endophyte communities of 67 host individuals in six epiphytic bromeliad species differing in C$_3$ and crassulacean acid metabolism (CAM) photosynthetic pathways. We tested whether endophyte assemblages were associated with ecophysiological leaf traits related to host photosynthetic pathways. Our results indicate that (1) C$_3$ and CAM bromeliads host dissimilar endophyte assemblages, (2) endophyte communities in C$_3$ bromeliads are characterized by variable relative abundances of fungal orders; conversely, CAM associated endophyte communities were characterized by consistent relative abundances of fungal orders, and (3) endophyte communities in bromeliads are distributed along a continuum of leaf toughness and leaf water content. Taken together, our study suggests that host physiology and associated ecophysiological traits of epiphytic bromeliads may represent biotic filters for communities of fungal endophytes in the tropics.

**Keywords:** endophytes; bromeliads; crassulacean acid metabolism; ecophysiological leaf traits; photosynthetic pathways

## 1. Introduction

Although often overlooked, vascular epiphytes are a distinctive and integral component of tropical forests, contributing between 25% and 35% of plant diversity [1] and up to 35% of the biomass of flowering plants [2]. Vascular epiphytes include major taxonomic groups such as orchids, ferns, aroids, and bromeliads [3]. The family Bromeliaceae is a monophyletic group of flowering plants represented by 59 genera and some 2400 species native to the Neotropics [4]. In the humid tropics, bromeliads may be terrestrial but are more often epiphytic, living non-parasitically on other plants, and relying on the atmospheric deposition of water and nutrients for their survival [4,5]. What is more, tropical bromeliads form relationships with a wide variety of tropical microbiota.

Tropical bromeliads frequently form associations with an abundant and diverse group of microsymbionts. These can include, algae [6], freshwater protozoa [7], yeasts [8], phyllosphere bacteria [9], and arbuscular mycorrhizal fungi [10,11]. However, because of their cryptic nature, foliar fungal

endophytes—microfungi living within asymptomatic plant tissues—have seldom been studied in bromeliads. Research on fungal endophytes in bromeliads has primarily focused on fungi living within root tissues (i.e., dark septate endophytes) and their associations with plant functional traits [12,13], yet fungal endophytes within bromeliad leaf tissues have remain underexplored [14,15]. One study compared endophytic fungal communities between tropical trees (*Hevea* spp.) and epiphytic bromeliads (*Tillandsia* spp.) and found distinct communities [14]. The reasons for the different endophyte communities remain unknown but could be due to the fact that the sampled plants were from distinct geographic regions and had different photosynthetic pathways (*Hevea* spp. use the $C_3$ photosynthetic pathway and *Tillandsia* spp. use the water-conserving, crassulacean acid metabolism (CAM) photosynthetic pathway). Plants that vary in their photosynthetic pathway also exhibit distinct differences in leaf ecophysiological traits; CAM species often have larger cells to store malate and thicker leaves to reduce the $CO_2$ leakage relative to their $C_3$ counterparts [16]. Foliar fungal endophytes (hereafter, endophytes) are known to be highly abundant and hyperdiverse in ferns, grasses, and woody angiosperms and gymnosperms in the tropics [17–20]. Still, our understanding of how plant functional traits are associated with structuring endophyte abundance, diversity, and community composition in tropical plants remains limited to understory woody trees. For example, endophyte communities of trees in a lowland rainforest of Papua New Guinea were correlated with variation in foliar traits, including leaf mass per area (LMA) and leaf carbon and nitrogen, and multivariate analyses showed that community composition of endophytic fungi in 10 dominant trees species in a temperate rainforest in Southern Chile were associated with variation in leaf resistance traits, such as leaf toughness and leaf anthocyanins [21]. Epiphytic bromeliads in the rainforest canopy with distinct ecophysiological leaf traits, offer an intriguing plant–fungal system by which to investigate the relationship between host functional traits and foliar endophyte communities.

Epiphytic bromeliads live as either 'tank' or 'atmospheric' forms, with contrasting photosynthetic pathways and leaf characteristics [4]. Indeed, epiphytic bromeliads are an ideal group of plants with which to address questions about the relationship between host functional traits and endophyte communities. Epiphytic bromeliads in the canopy often exhibit CAM or the $C_3$ mode of carbon metabolism [22,23]. CAM bromeliads tend to occur in stressful, dry environments, and their leaves are generally thick and highly succulent, low in specific leaf area (SLA), have thick impermeable cuticles to reduce water loss, and have a high density of trichomes, which are used to absorb water and nutrients from the atmosphere (i.e., atmospheric bromeliads) [4,22]. In contrast, $C_3$ bromeliads tend to occur in less stressful, wet environments. Their leaves are often thin and less succulent, are arranged in a rosette to collect water and nutrients (i.e., tank bromeliads that form phytotelma), are high in SLA, have thin cuticles, and lack dense surface trichomes [22,24]. It has been hypothesized that variation in leaf traits may act as host-imposed habitat filters to endophyte colonization [25], and given that endophytes colonize and spend most of their life within leaf tissues, contrasting photosynthetic pathways and associated ecophysiological leaf traits of epiphytic bromeliads may create differing microhabitats that act as filters for specific endophyte communities, and thus are significant contributors in the structuring of foliar endophyte communities.

In this study, we examine the associations of foliar fungal endophyte communities to photosynthetic pathways and related ecophysiological leaf traits of six common epiphytic bromeliads in a tropical wet rainforest in Costa Rica. We hypothesize that endophyte communities will differ between epiphytic CAM and $C_3$ bromeliads, with host ecophysiological leaf traits being one component driving these differences. Our aim is to (i) test for differences in endophyte assemblages in $C_3$ and CAM epiphytic bromeliad species, (ii) characterize the foliar endophyte communities within CAM and $C_3$ species driving these differences, and (iii) test for associations between host ecophysiological leaf traits and endophyte community composition. To our knowledge, this is one of the first studies to directly test the relationships of foliar fungal endophytes with plant photosynthetic pathways and associated leaf functional traits.

## 2. Materials and Methods

This study was conducted at La Selva Biological Research Station, Heredia, Costa Rica (83°59′ W, 10°26′ N, 40 m a.s.l.). La Selva is located in Northeastern Costa Rica and includes 1600 ha of lowland tropical wet forest [26]. La Selva receives an annual precipitation of 4000 mm, primarily during the wet season, from May to January. Moreover, La Selva has an average monthly precipitation of 382 mm, and an average monthly temperature of 25.8 ± 0.2 °C that varies little throughout the year [27].

In June of 2016, we examined the foliar endophyte communities of epiphytic bromeliads located within the crowns of *Virola koschnyi* (Myristicaceae) trees. We chose *V. koschnyi* trees because they host a diverse and robust epiphytic bromeliad community and exhibit myristicaceous branching, whereby multiple branches radiate out perpendicular to the trunk. These branches exhibit steep gradients in microhabitats and microclimates (e.g., vapor pressure deficit, VPD; and light) from the inner to the outer crown that host different bromeliad species with varying functional traits [24]. This is important to control for variations in spore rain that are known to be influenced by height [24]. Additionally, we selected *V. koschnyi* trees in order to control for tree characteristics that could influence endophyte colonization patterns. We sampled bromeliads from nine *V. koschnyi* trees with a diameter at breast height (DBH) >70 cm that were safe to climb, and that had three or more species of bromeliads within the crown. The average distance between *V. koschnyi* trees was 0.97 km (range: 0.21–2.1 km)

We sampled leaves from six of the most abundant and common epiphytic bromeliad species, three of which use C$_3$ photosynthesis, and three that use CAM (Figure 1 and Table 1). We used single-rope climbing techniques to survey bromeliads on the lowest 5–6 branches. Within the crown of the tree, we collected three healthy, mature leaves per individual plant, sampling from two individual plants per species per tree. We collected leaves by detaching them from the base, wrapping them in a moist paper towel, and placing leaves in Ziploc bags. We processed leaves for endophyte isolation within 12 h of collection (see details below).

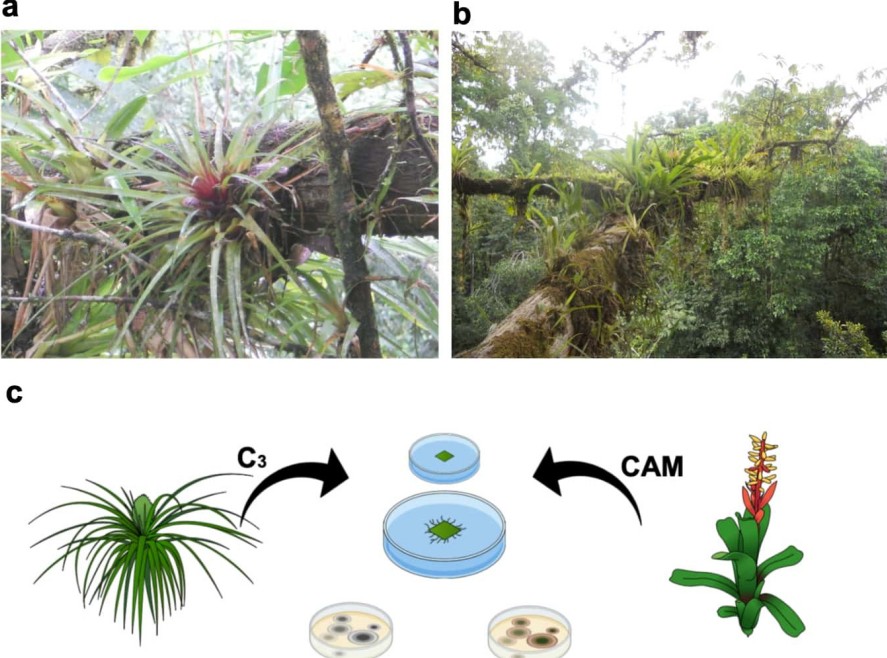

**Figure 1.** Six bromeliad species were sampled for endophytic fungi in Costa Rica. (**a**) *Tillandsia anceps* was one of our species that typifies the C$_3$ photosynthetic habit. (**b**) The C$_3$ and crassulacean acid metabolism (CAM) species in our study lived side-by-side on the same branches of *Virola kochsnyi*. (**c**) To study the fungal endophyte communities, we sampled leaf pieces from 3 C$_3$ and 3 CAM species. We grew the endophytes in agar and moved them into pure culture for DNA extraction, sequencing, and classification. Figure produced using mindthegraph.com.

**Table 1.** Epiphytic bromeliads surveyed from the forest canopy at La Selva Biological Station, Costa Rica. Code: species code; Photo: photosynthetic pathway; Form: morphological life form; N: number of bromeliads sampled and used for abundance and relative abundance estimates (number of bromeliads that produced ≥9 OTU and were used in multivariate analyses); Abundance: endophyte isolation frequency (mean ± SD); Richness: number of unique fungal operational taxonomic units (OTUs; mean ± SD); and Diversity: fungal diversity calculated as Fisher's Alpha (mean ± SD).

| Bromeliads | | | | | | Endophytes | | |
|---|---|---|---|---|---|---|---|---|
| **Genus** | **Species** | **Code** | **Photo** | **Form** | **N** | **Abundance** | **Richness** | **Diversity** |
| *Aechmea* | *A. nudicaulis* | AecNud | CAM [1] | Tank | 10 (7) | 0.71 ± 0.18 | 11.3 ± 3.16 | 14.3 ± 6.40 |
| *Tillandsia* | *T. bulbosa* | TilBul | CAM [1] | Atmospheric | 10 (8) | 0.78 ± 0.30 | 10.8 ± 4.39 | 11.9 ± 6.39 |
| *Tillandsia* | *T. festucoides* | TilFes | CAM [1] | Atmospheric | 9 (8) | 0.78 ± 0.24 | 12.8 ± 3.31 | 17.1 ± 13.5 |
| *Guzmania* | *G. monostachia* | GuzMon | $C_3$ [1] | Tank | 10 (3) | 0.34 ± 0.24 | 7.00 ± 4.42 | 15.6 ± 3.81 |
| *Tillandsia* | *T. anceps* | TilAnc | $C_3$ [1] | Tank | 11 (10) | 0.84 ± 0.20 | 11.7 ± 3.63 | 10.9 ± 7.78 |
| *Tillandsia* | *T. monadelpha* | TilMon | $C_3$ [1] | Tank | 13 (11) | 0.65 ± 0.25 | 11.0 ± 2.75 | 11.7 ± 7.54 |

[1] Tissue carbon isotope ratio ($\delta^{13}C$) was calculated to determine the occurrence of CAM and $C_3$ photosynthesis [28].

## 2.1. Host Ecophysiological Traits

We obtained host functional trait data for epiphytic bromeliads based on previously collected data at La Selva Biological Research Station [24] where two fully expanded leaves were selected without evidence of damage from six to ten adult individuals of each species found in six different *V. koschnyi* trees. These trees were not found to be significantly different in epiphyte species composition and microhabitats within their crowns as measured by Mooney et al. [29]. Leaf trait data included: specific leaf area (SLA; $mm^2$ $mg^{-1}$), fresh weight (g), dry weight (g), leaf succulence (g $m^{-2}$), leaf area ($mm^2$), leaf dry matter content (LDMC; mg/g), sclerophylly (g $mm^{-2}$), leaf resistance to fracture (N $mm^{-1}$), leaf toughness (N $mm^{-2}$), and rate of epidermal water loss (%RWC $h^{-1}$; Table 2, Figure S1).

## 2.2. Fungal Cultures

Prior to endophyte isolation, we washed each leaf in running tap water for 1 min, lightly cleaned off debris from the surface, and let the leaves air-dry. Next, we isolated fungi from leaves following [15]. For each of the three leaves per host individual, we removed the tips and margins of the leaf, and cut the remaining lamina into 4 $mm^2$ segments. For each individual plant, we pooled leaf segments into one sample, and surface-sterilized leaf segments by rinsing sequentially in 95% ethanol (10 s), 10% chlorine bleach (0.525% $NaOCl^-$; 2 min), and 70% ethanol (2 min). We allowed the leaf segments to surface-dry under sterile conditions before proceeding to the next step.

For each host sampled, we randomly selected 96 leaf segments and placed them individually into 1.5 mL microcentrifuge tubes containing 2% malt extract agar (2% MEA slants; 0.75 mL MEA/tube). We incubated slant tubes at room temperature and fungal growth began after 1 week, and isolates were left in slants for 5 months before transferring to pure culture. For each individual host, we isolated emergent fungi into pure culture on 2% MEA in Petri plates (35 mm diameter). We used the fungal cultures grown in plates for vouchering and DNA extraction, and deposited living vouchers in sterile water at Tulane University (Van Bael Lab, accession BP0001-BP1248).

**Table 2.** Description of select leaf functional traits from epiphytic bromeliads found in *Virola koschnyi* trees at La Selva Biological Station in Costa Rica. Adapted from Woods 2013 [24].

| Leaf Functional Trait | Formula | Units | Relations to Plant Performance |
|---|---|---|---|
| Specific leaf area (SLA) | Leaf area/dry weight | $mm^2 \, mg^{-1}$ | Correlates positively with growth rate and negatively with leaf life span [1]. |
| Succulence | (Fresh weight-dry weight)/leaf area | $g \, m^{-2}$ | Correlates with amount of water storage in plant tissue [2,3]. |
| Leaf resistance to fracture | Force/penetrometer circumference | $N \, mm^{-1}$ | Indicates carbon investment in structural protection; correlates positively with leaf life span [1] |
| Leaf toughness | (Force/penetrometer circumference)/leaf thickness | $N \, mm^{-2}$ | Correlates positively with leaf life span [4,5]. |
| Sclerophylly | Dry weight/leaf area | $g \, mm^{-2}$ | Correlates positively with leaf life span [5] |
| Rate of Epidermal water loss (EWL) | Δ%Relative water content/h | $\%RWC \, h^{-1}$ | Relates to cuticle thickness and is low in low water environments [3] |

[1] Cornelissen et al. [30]; [2] Mantovani [31]; [3] Lorenzo et al. [32]; [4] Wright and Cannon [33]; [5] Witkowski and Lamont [34].

### 2.3. DNA Extraction and Collections

We extracted DNA from fresh mycelium grown in culture using the Extract-N-Amp Plant PCR Kit (Sigma-Aldrich, Milwaukee, WI, USA) [35]. Nuclear ribosomal internal transcribed spacer regions (ITS1 and ITS2), 5.8S, and partial LSU (large ribosomal subunit) were amplified using the primers ITS1F and LR3 [36]. When primers ITS1F and LR3 failed to amplify the ITSrDNA-LSUrDNA region, we used primers ITS1F and ITS4 [37] to amplify only the ITSrDNA region. We verified PCR products using gel electrophoresis, and positive amplicons were cleaned and sequenced bidirectionally using the original primers at the GENEWIZ sequencing facility (South Plainfield, NJ, USA).

We used the ChromaSeq package in Mesquite v. 3.4 [38] and the programs phred and phrap to call bases and assemble contigs [39]. We manually edited contigs using Sequencher v. 5.1 (Gene codes Corporation, Ann Arbor, MI, USA). Overall, we successfully isolated and sequenced ITSrDNA or ITSrDNA-LSUrDNA for 1473 fungal cultures. All sequences are deposited in the NCBI Genbank under accession numbers MW045835–MW046031.

To visualize and determine the phylogenetic placement of isolates within the Pezizomycotina (Ascomycota), we used the tree-based alignment selector (T-BAS) toolkit [40]. Additionally, we used T-BAS to designate operational taxonomic units (OTUs) on the basis of 95%, 97%, and 99% sequence similarity. We used 97% sequence similarity to designate approximate species boundaries [37,41].

### 2.4. Statistics

We calculated endophyte abundance as the percent of leaf segments per 96 segments per host that produced fungal isolates in slant tubes for each of the 63 individual hosts. Next, we calculated endophyte diversity using Fisher's alpha on individual hosts that produced ≥9 distinct OTUs. Only 46 host individuals produced ≥9 unique OTUs, and these individuals were used for all subsequent multivariate statistical analyses. We tested for differences in endophyte abundance, richness, and diversity as a function of host species and photosynthetic pathways using Kruskal–Wallis tests.

We used unconstrained non-metric multidimensional scaling (NMDS) ordination to visualize endophyte community differences of Bray–Curtis dissimilarities among individual hosts. To test for significant differences in endophyte community composition among host species and host

photosynthetic pathways (i.e., C$_3$ vs. CAM) we used a permutational multivariate analysis of variance (PERMANOVA; 999 permutations) on the endophyte community matrix. Additionally, we used a permutational analysis of multivariate dispersion (PERMDISP) as a companion to PERMANOVA to ensure that differences observed were due to endophyte community composition and not endophyte community heterogeneity.

As suggested by Anderson and Willis [42], we used a constrained distance-based redundancy analysis (dbRDA) as a companion to NMDS to better view location differences among groups, given that group differences in multivariate space may not be apparent in unconstrained ordination. All models were based on Bray–Curtis dissimilarity and we used various functions from the R package vegan [43] to construct the dbRDA model. We began by determining the variance inflation factors of all predictor variables, using the vif.cca function. We then reduced the set of candidate variables, in a stepwise fashion, to a group that could completely test our hypotheses and had the lowest variance inflation values for the set. We used the ordiR2step function to perform forward model selection and then investigated variations on the results to give perspective to our conclusions. For all models we determined *p*-values for relationships between predictor variables and variation in community composition by performing Type III ANOVAs with the anova.cca function. Prior to constructing the model, leaf functional trait variables were mean-centered and scaled to unit variance.

## 3. Results

### 3.1. Endophyte Abundance, Richness, and Diversity

From 1521 fungal isolates in culture, we obtained a total of 1473 sequences (23.4 ± 6.56 per individual host; mean ± SD) across the entire dataset. From the 1473 sequences that could be placed phylogenetically, 1401 (95%) isolates were placed in the phylum Ascomycota, 48 (3%) isolates were placed in phylum Basidiomycota, and 24 isolates (2%) were placed in phylum Zoopagomycota and Mucoromycota. Within the phylum Ascomycota, which generally make up the foliar communities in tropical plants, we found that the classes Sordariomycetes comprised 95.4% of the foliar communities, while the next highest classes constituted Dothideomycetes, Arthoniomycetes, and Saccharomycetes (3.71%). After grouping sequences at 97% similarity, we obtained 223 unique OTUs, of which 144 (64.6%) occurred only once (singletons).

Endophyte abundance ranged from 0.20 to 1.00 (0.23 ± 0.50 per individual host; mean ± SD) and did not differ significantly as a function of photosynthetic pathway ($\chi^2$ = 3.41, df = 2, *p* = 0.21) or host species ($\chi^2$ = 2.63, df = 6, *p* = 0.43). After removing individual hosts with less than nine distinct OTUs, OTU richness ranged from 9 to 20 OTUs (12.6 ± 2.85 per individual host; mean ± SD). We did not find any differences in endophyte richness as a function of photosynthetic pathway ($\chi^2$ = 1.53, df = 2, *p* = 0.46) or host species ($\chi^2$ = 2.45, df = 6, *p* = 0.87). Over the entire dataset, Fishers Alpha ranged from 4.55 to 38.7 (13.1 ± 8.3 per individual host; mean ± SD). We did not find significant differences in endophyte diversity as a function of photosynthetic pathway ($\chi^2$ = 1.35, df = 2, *p* = 0.51) or host species ($\chi^2$ = 4.67, df = 6, *p* = 0.58).

### 3.2. Endophyte Communities and Host Photosynthetic Pathways

The NMDS ordination revealed that endophyte community composition differed significantly among bromeliads as a function of host photosynthetic pathway (Figure 2; PERMANOVA: F = 4.92, *p* < 0.001, PERMDISP: F = 2.54. *p* = 0.13). Additionally, endophyte communities differed significantly by host species, although with some overlap in fungal communities (Figure 2, PERMANOVA: F = 2.35, *p* < 0.001; PERMDISP: F = 2.84, *p* = 0.14). Host species and photosynthetic pathways each explained 22.3% and 9.85% of the variation in endophyte community composition, respectively. When controlling for photosynthetic pathways, host species accounted for just 12.5% of the variation in endophyte community composition.

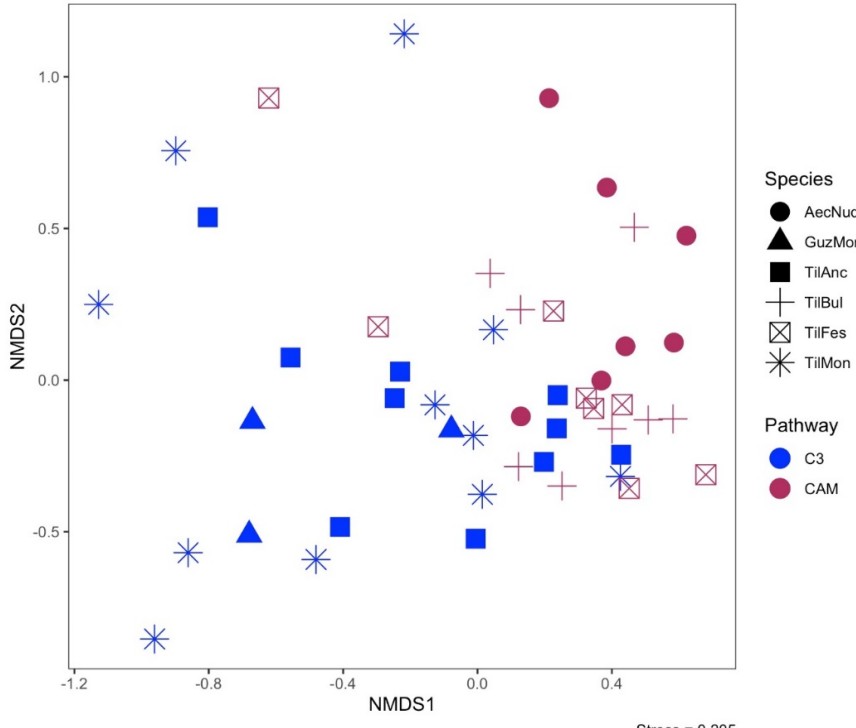

**Figure 2.** Nonmetric multidimensional dimensional scaling (NMDS) ordination of fungal endophyte communities among bromeliads species differing in CAM and $C_3$ photosynthetic pathways. Each point represents an endophyte community sampled from an individual bromeliad (*n* = 47).

*3.3. Endophyte Orders within Bromeliad Species*

A heat map of the relative abundance of fungal orders across the entire data set revealed that Xylariales, Sordariales, and Hypocreales were the three most abundant orders across the six bromeliads species (Figure 3). Xylariales were the most abundant fungi in CAM bromeliads (range: 40–49%) followed by Sordariales (26–34%). Xylariales and Sordariales showed a more heterogeneous distribution within the $C_3$ bromeliads; Xylariales were most abundant in *Tillandsia anceps* (38%), and Sordariales were most abundant in *T. monodelpha* (29%). Hypocreales fungi were generally found within $C_3$ bromeliads (27–64%), and were highly abundant in *Guzmania monostachia* (64%); in contrast, Hypocreales were found at low abundances (8–11%) in CAM bromeliads. The relative abundance of Hypocreales varied less within CAM bromeliads compared to $C_3$ bromeliads (Figure S2), though no significant test was performed. On average, the relative abundances of Xylariales and Sordariales were higher in CAM, relative to $C_3$ bromeliads, while Hypocreales were higher in $C_3$ bromeliads.

*3.4. Endophyte Communities and Host Ecophysiological Traits*

Endophyte communities in CAM were significantly associated with photo system and mechanical and structural leaf traits related to water conservation and best explained by the dbRDA model: Community composition-PhotoSystem + Leaf Fracture (Figure S3).

However, to elucidate relationships beyond what had already been concluded by the NMDS and PERMANOVA (Figure 2), we selected another model that ignored the photo system and found sclerophylly and a leaf fracture to explain the most variation in community composition when the model was blind to the photo system. (Figure 4, Table 3). Leaf resistance to the fracture correlated with endophyte communities found in the CAM bromeliads, *Aechmea nudicaulis*. Leaf sclerophylly correlated with endophyte communities in CAM bromeliads, *T. festucoides* and *T. bulbosa*, and were tightly clustered when constrained by leaf sclerophylly (Figure 4). Endophyte communities within

the C$_3$ bromeliads were negatively associated with leaf sclerophylly, and showed a wider distribution along the sclerophylly spectrum (Figure 4).

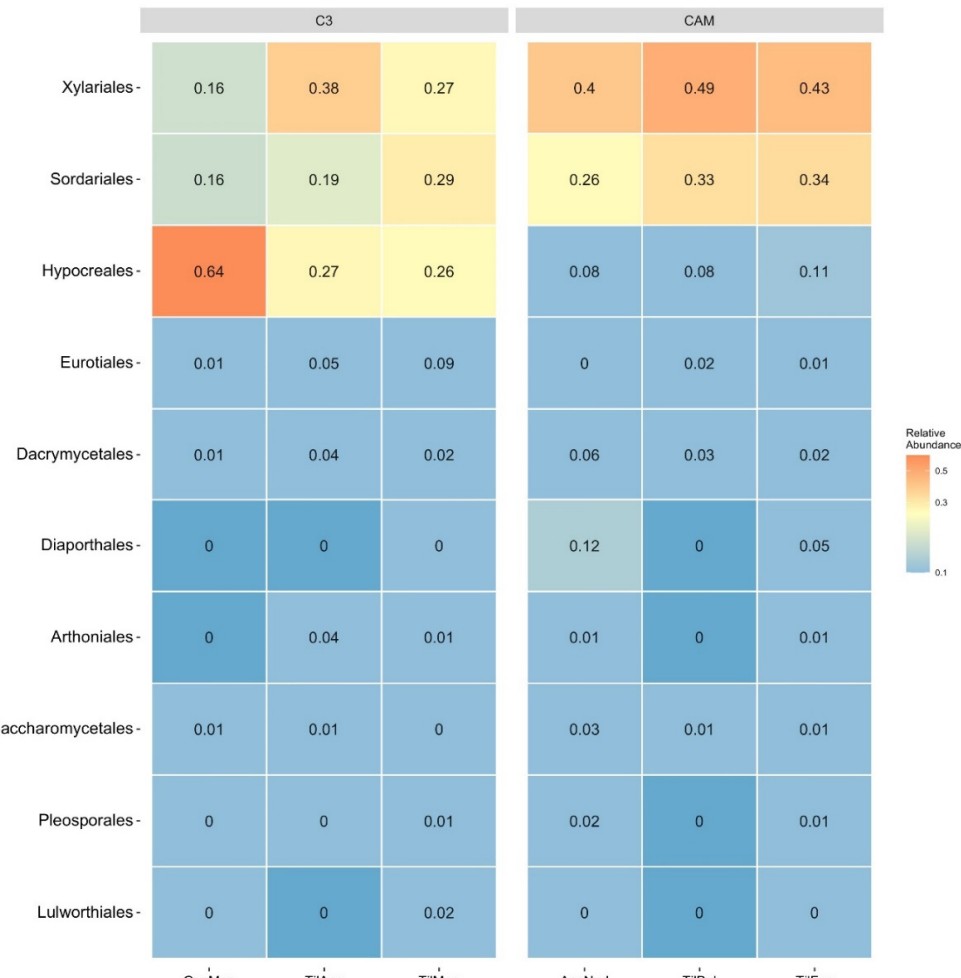

**Figure 3.** Heat-map of percent relative abundance of fungal orders (left) by host species (bottom); for a list of abbreviations, see (Table 1) and faceted by photosynthetic pathway (top) from the culture-based survey. Numbers within the map are the mean percent relative abundance of fungal orders within a host species.

We also investigated a model that depended solely on the photosynthetic pathway, dry weight, and SLA, to compare the explanatory power of these seminal leaf traits with those that were determined through model selection but found the model to have little explanatory power compared with the above models (Table S1). It is worth noting that *T. festucoides* and *A. nudicaulis* had extremely large values with regard to SLA and succulence values, respectively. We tested the effects of excluding these species and found the resultant models explained less variation in community composition than the models described above.

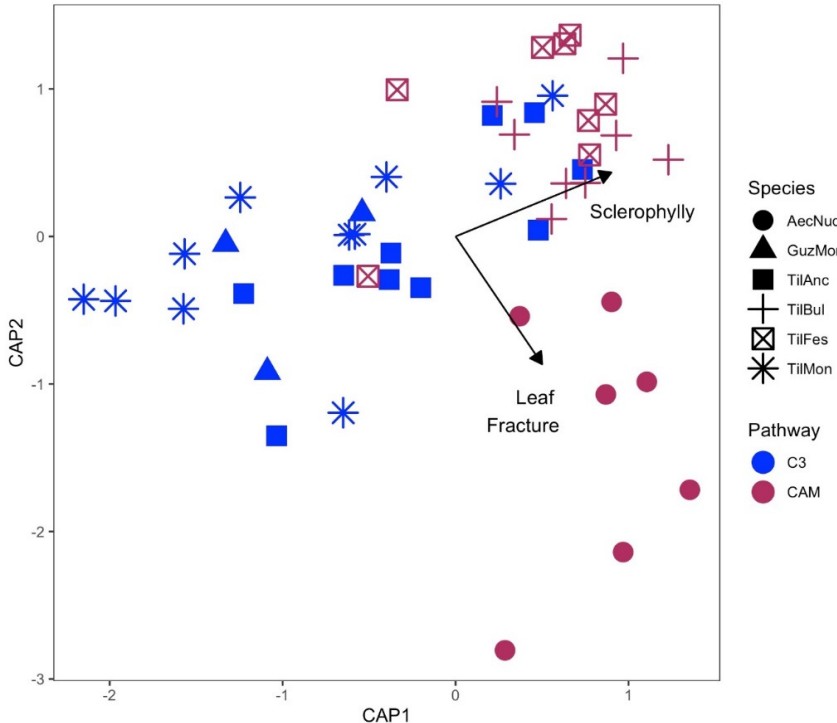

**Figure 4.** Distance-based redundancy analysis (dbRDA) ordination of leaf traits associated with endophyte communities of CAM and $C_3$ bromeliad species. Arrows represent significant positive associations between host functional traits and endophyte communities, and the length signifies the magnitude of the association ($n = 47$).

**Table 3.** PERMANOVA marginal test. Type III ANOVA-like permutational test. Associations between bromeliad functional traits and endophyte community composition (Bray–Curtis) constrained in a dbRDA model.

| Model | Leaf Functional Traits | F | *p*-Value |
|---|---|---|---|
| Model 1 | Photosynthetic pathway | 4.09 | 0.001 |
| | Leaf resistance to fracture | 2.37 | 0.003 |
| Model 2 | Sclerophylly | 4.430 | 0.001 |
| | Leaf resistance to fracture | 2.837 | 0.002 |

## 4. Discussion

Our study explored the relationship between foliar fungal endophytes and epiphytic bromeliads in a lowland tropical rainforest. We focused on fungal endophyte communities and their relationships to host photosynthetic pathways and associated ecophysiological leaf traits. Studies of endophyte communities associated with host ecophysiological traits are lacking, therefore, we highlighted three key findings from our study. First, variation in endophyte community composition differed between CAM and $C_3$ bromeliads. Second, endophyte assemblages in $C_3$ bromeliads were characterized by variable fungal order abundances; conversely, CAM associated endophyte assemblages were characterized by consistent relative abundances of fungal orders. Last, the key leaf functional traits related to water retention and host defense strategies were associated with foliar endophyte communities. We discussed possible implications of our results.

### 4.1. CAM vs. $C_3$ Endophyte Communities

We found that the endophyte assemblages differed between $C_3$ and CAM bromeliads (Figure 2). We posited that the differences in endophyte communities between CAM and $C_3$ bromeliads might

be due to the contrasting ecophysiological properties of CAM and $C_3$ bromeliads, which presumably present differing microenvironments for fungal colonization and specialization. For example, CAM bromeliads generally tend to have reduced water loss due to daytime stomatal closure [3], and higher water use efficiency (the ratio of water use in plant metabolism to water lost through transpiration) [44], relative to $C_3$ bromeliads. They also exhibit superior $CO_2$ acquisition during the wet season [45], and recycling of respiratory $CO_2$ under stress (i.e., CAM cycling or CAM idling [44]). These conditions may result in a stable microenvironment for endophytes by maintaining photosynthetic efficiency (and carbohydrate levels, which presumably endophytes consume), water levels, and internal leaf conditions constant for CAM plants. Indeed, the relative abundances of fungal orders Xylariales and Sordariales remained consistent across CAM bromeliads (Figure 3).

In contrast to CAM bromeliads, $C_3$ bromeliads may present a more variable microenvironment for endophyte communities, given that $C_3$ plants generally do not possess the physiological plasticity of CAM bromeliads, which are adapted to the arid microclimate of the tree canopy [22]. We hypothesized that a variable microenvironment within the different $C_3$ bromeliad species may select for varying endophyte assemblages that are dependent on host ecophysiology. Evidence for this can be seen in the inconsistent relative abundances of fungal orders Xylariales, Sordariales, and Hypocreales across $C_3$ bromeliads, compared to CAM bromeliads (Figure 3). Future studies focusing on a broader range of CAM and $C_3$ bromeliad species will provide clearer patterns of endophyte community affiliations with photosynthetic pathways.

### 4.2. Endophyte Orders in Bromeliad Species

Over the entire dataset, we found a diverse community of fungal endophytes within epiphytic bromeliads, composed primarily of fungal orders Xylariales, Sordariales, and Hypocreales (Figure 3). Xylariales were abundant across the six bromeliad species, having consistently higher abundances in CAM relative to $C_3$ bromeliads. Xylariales are a phylogenetically diverse and ubiquitous group of fungi in the tropics, and as endophytes, they are the most commonly isolated order found in tropical plants, including woody plant species [19,36], forest grasses [18], ferns [17], and epiphytic bromeliads [14]. Our results are consistent with studies showing Xylariales as the most common order of fungi, alongside Sordariales, another common abundant fungal group found in tropical plants [19].

Sordariales endophytes were found across all bromeliad species but were particularly abundant in CAM bromeliads, relative to $C_3$ bromeliads. A culture-based study of endophytes (OTU based on 97% sequence similarity) within epiphytic bromeliads in the Peruvian highlands found that Sordariales and Xylariales were predominantly isolated from epiphytic CAM bromeliads, *Tillandsia usneoides*, *T.* cf. *purpurea*, and *T.* cf. *cacticola*, compared to tropical terrestrial woody plants *Hevea* spp. and *Vasconcellea microcarpa* [14]. The same study found the Hypocreales fungi were relatively absent in the same CAM *Tillandsia* spp., with only two fungal OTUs found. Similarly, our study found that Hypocreales fungi were poorly represented in the CAM bromeliads *T. bulbosa*, *T. festucoides*, and *A. nudicaulis*, and were in higher abundances in the $C_3$ bromeliads, *T. anceps*, *T. monadelpha*, and *G. monostachia*, suggesting that Sordariales, but not Hypocreales, are adapted to the physiological microenvironment of CAM bromeliads. $C_3$ bromeliads in our study exhibit a 'tank form'—overlapping leaves that impound water known as a phytotelma—and as a result, create a pool habitat for a community dominated by arthropods (primarily insects and arachnids in various stages of development) [4,22]. In addition to plant pathogens and mycoparasites, the order Hypocreales contains the largest number of entomopathogenic fungi with a wide range of invertebrate hosts [46], and previous biological control work has shown that entomopathogenic fungi can also be isolated as endophytic fungi from plants [47]. We hypothesized that the higher abundances of Hypocreales in $C_3$ tank-forming bromeliads may be related to the abundance of microinvertebrates living in the water-collecting phytotelma of $C_3$ bromeliads, but future studies are needed to test this hypothesis by growing bromeliads in a controlled environment with and without invertebrates.

### 4.3. Leaf Functional Traits and Endophyte Communities

When we constrained the endophyte communities by leaf traits, we found that endophyte communities in bromeliads were distributed along a continuum of leaf traits associated with water retention and leaf defense strategies (Figure 4). Sclerophylly is a textural form of a leaf—described as rough, stiff, and hard leathery leaves [48] at one end of the leaf economic spectrum [49], and it has been suggested to be an adaptation to seasonal drought, with thick cuticle and leaves contributing to water conservation [50]. Endophyte communities within CAM bromeliads, *T. festucoides* and *T. bulbosa*, were more similar in composition when constrained by leaf sclerophylly, with tighter clustering around a high-water conservation strategy (i.e., high sclerophylly). In sharp contrast, endophyte communities of $C_3$ bromeliads, *T. anceps*, *T. monadelpha*, and *G. monostachia,* had a wider distribution along the water conservation spectrum (i.e., low to high leaf sclerophylly), suggesting a broader range in microhabitat.

Leaf resistance to fracture is a mechanical leaf trait often associated with protection against herbivores and pathogens in tropical woody plants [48], but epiphytic bromeliads have lower rates of attack by natural enemies due to their low foliar nutrient content [51]. Instead, epiphytic bromeliads are subjected to harsh environmental conditions in the forest canopy and leaf resistance to fracture may be related to maintaining leaf integrity to form the phytotelma [24]. Therefore, associations of endophyte communities in *A. nudicaulis* with leaf resistance to fracture may be a result of another mechanical trait associated with water retention. Furthermore, leaf resistance traits have been shown to be associated with fungal endophyte communities in tropical woody plants [21], and it is hypothesized that leaf resistance to fracture may limit endophyte colonization, given that some colonization occurs via penetration pegs through the surface of leaves, and leaf resistance to fracture may constitute a physical barrier against endophyte colonization [52]. Tougher leaves may prevent certain endophyte taxa from colonizing, while less tough leaves would allow for easier endophyte entry, consequently affecting community structure. Given that leaf fracture toughness and sclerophylly are often associated with each other, and with limiting water loss [48], our results suggest that endophyte assemblages in the forest canopy may be structured by the water conservation strategies of their epiphytic hosts, dependent on the interplay of biomechanical and textural leaf properties of epiphytic CAM and $C_3$ bromeliads.

Our results are in line with recent studies showing associations between endophyte communities and leaf functional traits in the tropics. Gonzalez-Teuber et al. 2020 found that endophyte communities were related to the variation in leaf resistance traits such as the cell wall, flavonoids, anthocyanins, and terpenoids in 10 tree species from a temperate rainforest in Southern Chile [21]. Likewise, Vincent et al. 2016 found that the distribution of endophyte communities were correlated with leaf mass per area and leaf nitrogen and carbon of 11 tree species in a lowland wet rainforest in Papua New Guinea [20]. Our study added photosynthetic pathways, leaf sclerophylly and fracture toughness to a growing body of work linking leaf traits to endophyte community composition.

## 5. Conclusions

Our study is the first to highlight the importance of photosynthetic pathways and associated ecophysiological leaf traits as potential drivers of fungal endophyte communities in tropical epiphytic plants. Several factors are known to influence endophyte communities: dispersal limitation [18], geographical position [37], structure and diversity of surrounding vegetation [53], and host identity [17], yet host-imposed habitat filters remain relatively under-explored (but see Saunders et al. [25]). Our study extends our understanding to include ecophysiological leaf traits as factors mitigating endophyte composition in the tropics. Foliar endophyte studies in the tropics have focused on the plants in the humid understory such as ferns [17], grasses [15], and woody angiosperms and gymnosperms [14,20,36,54], and the few studies of fungal symbionts of epiphytic bromeliads have focused primarily on mycorrhizal and dark septate endophyte symbionts [12,13], with few studies examining the foliar endophyte communities. Considering that vascular epiphytes compose up to 35% of the floral diversity and foliar biomass in tropical forests [2], our study addresses a knowledge

gap in foliar endophyte diversity measures in epiphytic bromeliads. Finally, although photosynthetic pathways and associated leaf traits may be one of several factors affecting endophyte communities in tropical plants, additional features of the epiphytic environment–epiphyte community structure, gradients of light, relative humidity within the tree canopy, canopy soil, and spore inoculum [24,29] should not be discounted. However, analyses of these factors were beyond the scope of this study. Our study constitutes a baseline for future studies to compare against and broadens our understanding of host-imposed drivers for foliar endophyte communities in the tropics. Further studies on bromeliads and their fungal partners would be desirable to provide a more complete evolutionary picture.

**Supplementary Materials:** The following are available online at http://www.mdpi.com/1424-2818/12/10/378/s1, Figure S1: Correlogram of relationships among bromeliad ecophysiological traits. Figure S2: Variation in relative abundance of common fungal orders in epiphytic bromeliads. Figure S3: dbRDA ordination of fungal endophytes associated with host ecophysiological traits. Table S1: ANOVA-like tests on dbRDA models of endophyte community composition and host predictive variables.

**Author Contributions:** Conceptualization, P.H.T., C.L.W., S.A.V.B.; methodology, P.H.T., C.L.W., S.A.V.B.; statistical analysis P.H.T., S.F.; investigation, P.H.T., C.L.W., S.A.V.B.; resources, P.H.T., C.L.W., S.A.V.B.; data curation, P.H.T.; writing—original draft preparation, P.H.T.; writing—review and editing, P.H.T., C.L.W., S.F., S.A.V.B.; visualization, P.H.T., S.F.; supervision, S.A.V.B.; project administration, P.H.T.; funding acquisition, P.H.T., S.A.V.B. All authors have read and agreed to the published version of the manuscript.

**Funding:** This research was funded by Organization for Tropical Studies, and Tulane's Ecology and Evolutionary Biology Department, Funding to S.A.V.B. from the Tulane School of Science and Engineering. This research was funded by National Science Foundation DEB-1556583 to S.A.V.B.

**Acknowledgments:** Field assistant Rigoberto Gonzales, and Elizabeth A. Arnold and Liz Kimbrough for comments on the MS.

**Conflicts of Interest:** The authors declare no conflict of interest.

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
