# Peer review of "Relationships between Foliar Fungal Endophyte Communities and Ecophysiological Traits of CAM and C3 Epiphytic Bromeliads in a Neotropical Rainforest"

_diversity, doi:10.3390/d12100378_

Round 1

Reviewer 1 Report

Relationship between foliar fungal endophyte communities and ecophysiological traits of CAM and C3 epiphytic bromeliads in a neotropical rainforest

The manuscript investigates the cultured fungal endophytic communities from CAM and C3 bromeliads and investigates if common plant traits (some associated with photosynthetic pathway) affect community assemblage. The manuscript is interesting and written in a slightly less formal style that is refreshing making it very readable. The scientific question is interesting and the results are not unexpected since similar habitats contain similar fungal communities.

Please address the following concerns within the manuscript for greater clarity.

1) Sterilized pieces of leaves were placed into 1.5 ml micro centrifuge tubes for isolation and fungal colonies were then transferred to Petri plates. If this is correct, was only one fungus isolated from each tube? What if there were more than one fungus in a tube, were both isolated? This is important because many fungi can be isolated from leaf tissue, some grow fast and others slow. The isolation area within a 1.5 ml tube is small and may limit the ability to isolate several fungi from the same leaf tissue. How was this addressed and how might this alter the results?

2) The conclusion section should be reorganized and reduced to one paragraph.

Specific suggestions:

Line 27 and 28: ... and leaf succulence.

Suggest changing succulence to water content for word consistency.

Line 51: use the C3 ...

Suggest changing 3 to subscript for consistency.

Line 52: ... crassulacean acid metabolism or CAM ...

This is the first time used so consider revising to:

... crassulacean acid metabolism (CAM) ...

Line 60: ... including LMA ...

Unless I missed it, LMA has not been defined within the manuscript so please write out.

Line 62: ... were associated in variation ...

Suggest changing in to with

Line 69: ... in the canopy often exhibit crassulacean acid metabolism (CAM)

Consider changing to the following:

... in the canopy often exhibit CAM ...

Line 73: ... wet environments and their leaves ...

Consider revising to two sentences:

... wet environments. Their leaves

Line 100: ... VPD ...

Unless I missed it, VPD has not been defined within the manuscript so please write out.

Line 101: ... and my dissertation).

Consider removing "and my dissertation" or use proper citation for the dissertation.

Figure1 C: Based on the written methods, this figure is a little misleading. Sterilized pieces of leaves were placed into 1.5 ml micro centrifuge tubes for isolation and fungal colonies were then transferred individually to Petri plates. The figure shows isolation of fungi on plates and individual colonies transferred to separate Petri plates.

Line 145: ... we haphazardly collected 96 ...

Consider revising:

... we randomly selected 96 ...

Line 154: ... spacer region and 5.8S gene and ca. 600bp..

Consider revising to be more specific because there are two spacer regions (ITS1 and ITS2).

Something like: Nuclear ribosomal internal transcribed spacer regions (ITS1 and ITS2), 5.8S, and partial LSU (large ribosomal subunit) were amplified using the primers ITS1F and LR3 (need references for the primers).

Line 215: ... in fungal communities. (...

Please remove the extra period after communities

Line 224: ... Sordariales, Hypocreales ...

Consider added "and" after the ,

... Sordariales, and Hypocreales ...

Table 3: ... Type III and dbRDA model.

Type and dbRDA are underlined, should they be?

Line 255: ... on photosynthetic pathway. dry weight, and SLA, to compare....

Consider revising:

... on photosynthetic pathway, dry weight, and SLA to compare

Line 282-283: ... suggesting an adaptation to a stable microenvironment.

I'm not sure the results support this conclusion. There are many reasons that one may see consistent colonization vs. variable colonization. Please remove. I think the sentences in lines 288-291 is more appropriate based on the experimental design.

Line 302: bromeliads. relative to C3 bromeliads.

Please remove the extra period.

Lines 342-345: This is only one sentence. Please move it up to the bottom of the previous paragraph.

Author Response

 Reviewer 1

1) Sterilized pieces of leaves were placed into 1.5 ml micro centrifuge tubes for isolation and fungal colonies were then transferred to Petri plates. If this is correct, was only one fungus isolated from each tube? What if there were more than one fungus in a tube, were both isolated? This is important because many fungi can be isolated from leaf tissue, some grow fast and others slow. The isolation area within a 1.5 ml tube is small and may limit the ability to isolate several fungi from the same leaf tissue. How was this addressed and how might this alter the results?

Response: Fungal endophytes in tropical plants frequently occur in densities approaching one endophyte isolate per each 2 mm2 pf mature leaf tissues [1]. In the case that one tube produced two or more fungal isolates, two isolates would be cultured unto plates and named sample# a, b, c, etc. This culturing method is a commonly used method for fungal isolations from leaf tissues [1-4] and we do not predict altering of the results due to this.

2) The conclusion section should be reorganized and reduced to one paragraph.

Thank you for this comment. The conclusion section has been reorganized and reduced to the one paragraph.

Specific suggestions:

Line 27 and 28: ... and leaf succulence.

Suggest changing succulence to water content for word consistency.

Lines 27-28: The text has been updated to change wording to succulence for word consistency.

Line 51: use the C3 ...

Suggest changing 3 to subscript for consistency

                Line 51: Text has been updated to reflect changing 3 to subscript.

Line 52: ... crassulacean acid metabolism or CAM ...

This is the first time used so consider revising to:

... crassulacean acid metabolism (CAM) ...

                Line 52: Text has been updated, revising to ‘crassulacean acid metabolism (CAM)’.

Line 60: ... including LMA ...

Unless I missed it, LMA has not been defined within the manuscript so please write out.

                Lines 60: Text has been updated to define LMA.

Line 62: ... were associated in variation ...

Suggest changing in to with

                Line 62: Text has been updated to reflect change from in to with.

Line 69: ... in the canopy often exhibit crassulacean acid metabolism (CAM)

Consider changing to the following:

... in the canopy often exhibit CAM ...

                Line 69: Text has been updated to ‘in the canopy often exhibit CAM’.

Line 73: ... wet environments and their leaves ...

Consider revising to two sentences:

... wet environments. Their leaves

                Line 73: Text has been updated to split sentence in to two sentences.

Line 100: ... VPD ...

Unless I missed it, VPD has not been defined within the manuscript so please write out

                Line 100: Text has been updated to define vapor pressure deficit.

Line 101: ... and my dissertation).

Consider removing "and my dissertation" or use proper citation for the dissertation

                Text has been added to use a proper citation.

Figure1 C: Based on the written methods, this figure is a little misleading. Sterilized pieces of leaves were placed into 1.5 ml micro centrifuge tubes for isolation and fungal colonies were then transferred individually to Petri plates. The figure shows isolation of fungi on plates and individual colonies transferred to separate Petri plates.

Response: Constraints in image quality and resolution made it difficult for the artist to show that sterilized pieces of leaves were in micro-centrifuge tubes. The decision was made to express a general conceptual workflow for how fungal endophytes were isolated.

Line 145: ... we haphazardly collected 96 ...

Consider revising:

... we randomly selected 96 ...

                Line 145: Text has been updated to reflect change to ‘we randomly selected 96’.

Line 154: ... spacer region and 5.8S gene and ca. 600bp..

Consider revising to be more specific because there are two spacer regions (ITS1 and ITS2).

Something like: Nuclear ribosomal internal transcribed spacer regions (ITS1 and ITS2), 5.8S, and partial LSU (large ribosomal subunit) were amplified using the primers ITS1F and LR3 (need references for the primers).

                Lines 154-157: Text has been updated to reflect change to recommended reviewer comment.

Line 215: ... in fungal communities. (...

Please remove the extra period after communities

                Line 216: Extra period has been removed after communities.

Line 224: ... Sordariales, Hypocreales ...

Consider added "and" after the ,

... Sordariales, and Hypocreales ...

                Line 225: ‘and’ has been added to text.

Table 3: ... Type III and dbRDA model.

Type and dbRDA are underlined, should they be?

                Line 255: Text has been updated to remove underlined word.

Line 255: ... on photosynthetic pathway. dry weight, and SLA, to compare....

Consider revising:

... on photosynthetic pathway, dry weight, and SLA to compare

Line 256: Period punctuation has been removed and replaced by a comma, after photosynthetic pathway.

Line 282-283: ... suggesting an adaptation to a stable microenvironment.

I'm not sure the results support this conclusion. There are many reasons that one may see consistent colonization vs. variable colonization. Please remove. I think the sentences in lines 288-291 is more appropriate based on the experimental design.

Line 283: Text has been updated to reflect the removal of ‘suggesting an adaptation to a stable microenvironment’.

Line 302: bromeliads. relative to C3 bromeliads.

Please remove the extra period.

                Line 302: Extra period has been removed.           

Lines 342-345: This is only one sentence. Please move it up to the bottom of the previous paragraph.

                Line 341: The sentence has been moved to the bottom of the previous paragraph.

  1. Arnold, A.E.; Mejía, L.C.; Kyllo, D.; Rojas, E.I.; Maynard, Z.; Robbins, N.; Herre, E.A. Fungal endophytes limit pathogen damage in a tropical tree. Proceedings of the National Academy of Sciences 2003, 100, 15649-15654, doi:10.1073/pnas.2533483100.
  2. Higgins, K.L.; Coley, P.D.; Kursar, T.A.; Arnold, A.E. Culturing and direct PCR suggest prevalent host generalism among diverse fungal endophytes of tropical forest grasses. Mycologia 2011, 103, 247-260.
  3. Del Olmo-Ruiz, M.; Arnold, A.E. Interannual variation and host affiliations of endophytic fungi associated with ferns at La Selva, Costa Rica. Mycologia 2014, 106, 8-21, doi:10.3852/13-098.
  4. Rojas-Jimenez, K.; Hernandez, M.; Blanco, J.; Vargas, L.D.; Acosta-Vargas, L.G.; Tamayo, G. Richness of cultivable endophytic fungi along an altitudinal gradient in wet forests of Costa Rica. Fungal Ecology 2016, 20, 124-131.

Reviewer 2 Report

This very well written manuscript deals with some ecological aspects concerning association of endophytic fungi with plants in the Bromeliaceae in a forest context of Central America, introducing some original points. However, a general positive judgment is counterpoised by a basic perplexity concerning the choice by the authors to discuss their findings with reference to the order, which in my opinion makes their considerations less significant. In fact, orders represent a taxonomic level grouping diverse species playing very diverse ecological roles, and it makes little sense treating them as a unique entity. As an example, the discussion at lines 310-318 concerning Hypocreales is quite misleading, considering that besides many entomopathogens this order includes important plant pathogens, Fusaria above all, and mycoparasites such as Trichoderma. Moreover, at some extent taxonomy of fungi is still unstable, and rearrangements due to ongoing phylogenetic assessments might introduce further adjustments in the placement of some taxa, with an ensuing impact on authors' considerations. Apart of these objections, it is a pity that information concerning species, as resulting from a huge work carried out by means of DNA-sequencing, is disregarded. Indeed it could be of preminent relevance, within a work which is basically descriptive of fungal diversity in such a peculiar ecological context.

In connection with the above taxonomic controversy, a correction is to be done at line 198. In fact, Zygomycota is no more accepted as a phylum name, and fungi previously classified in this taxon are now divided between the phyla Zoopagomycota and Mucoromycota, with the latter including Glomeromycota (Spatafora et al 2016). Corrections are also required at line 200 ('Sordariomycetes') and 201 ('Dothideomycetes').

In methodology, authors should clarify end of sentence at lines 129-130 ('as measured [28]'), and confirm correctness of duration of incubation (5 months, line 147). In fact most fungi start growing on the isolation media within a few days; did the authors obtain some isolates after such long time? Other minor corrections follow: delete 'wet' at line 82 (never heard of a dry rainforest...); change to 'at 97% similarity,' at line 201; 'function of' at line 213; 'although' at line 215; 'heterogeneous' at line 226; 'distribution' at line 248; 'pathway,' at line 255; 'Hevea' at line 305; 'suggest' at line 343.

Moreover, I propose to change line 127 like this '.. at La Selva Biological Research Station [23], where two fully expanded leaves were selected ..', considering that author's name is part of ref. [23]. For the same reason, delete 'Higgins et al., 2011' at line 139. Ref. [15] is inappropriate for statement at line 49? Finally, abbreviated journal names have to be used throughout the reference list.

Author Response

Reviewer 2

This very well written manuscript deals with some ecological aspects concerning association of endophytic fungi with plants in the Bromeliaceae in a forest context of Central America, introducing some original points. However, a general positive judgment is counterpoised by a basic perplexity concerning the choice by the authors to discuss their findings with reference to the order, which in my opinion makes their considerations less significant. In fact, orders represent a taxonomic level grouping diverse species playing very diverse ecological roles, and it makes little sense treating them as a unique entity. As an example, the discussion at lines 310-318 concerning Hypocreales is quite misleading, considering that besides many entomopathogens this order includes important plant pathogens, Fusaria above all, and mycoparasites such as Trichoderma. Moreover, at some extent taxonomy of fungi is still unstable, and rearrangements due to ongoing phylogenetic assessments might introduce further adjustments in the placement of some taxa, with an ensuing impact on authors' considerations. Apart of these objections, it is a pity that information concerning species, as resulting from a huge work carried out by means of DNA-sequencing, is disregarded. Indeed it could be of preminent relevance, within a work which is basically descriptive of fungal diversity in such a peculiar ecological context.

Response: We agree that with the reviewer that orders represent a taxonomic level grouping diverse species playing very diverse ecological roles. We focused on fungal orders so as to compare our results to results of similar studies on tropical fungal communities [1-5] which also use fungal orders to describe the fungal communities. This conservative method of focusing on order is consistent with the reviewer's comment that ongoing phylogenetic assessments are likely to place taxa in different families or orders in the future. Also, changing the analysis to focus on family or genera, for example, would not likely change the overall finding that the different photosynthetic pathways host unique communities. We did address the reviewer's comment about the discussion from lines 310-318 (now lines 358-360). To be more inclusive, we added the phrase, "In addition to plant pathogens and mycoparasites, the order Hypocreales...."

In connection with the above taxonomic controversy, a correction is to be done at line 198. In fact, Zygomycota is no more accepted as a phylum name, and fungi previously classified in this taxon are now divided between the phyla Zoopagomycota and Mucoromycota, with the latter including Glomeromycota (Spatafora et al 2016). Corrections are also required at line 200 ('Sordariomycetes') and 201 ('Dothideomycetes').

Lines 199-202: Text has been updated to reflect the change in fungal classification to phyla Zoopagomycota and Mucoromycota. Moreover, Corrections have been made to Sordariomycetes and Dothidiomycetes.

In methodology, authors should clarify end of sentence at lines 129-130 ('as measured [28]'),

                Line 130: Text has been updated to clarify citation.

and confirm correctness of duration of incubation (5 months, line 147). In fact most fungi start growing on the isolation media within a few days; did the authors obtain some isolates after such long time?

Line 147-148: Text has been upgraded to reflect that fungi were seen to growing after a week, and they were left incubated for 5 months before transferring to pure culture plates. 

Other minor corrections follow:

delete 'wet' at line 82 (never heard of a dry rainforest...);

Response: Tropical dry rainforests are an important component in tropical regions [6], and we wanted to differentiate our study site given that is defined as tropical wet forest. Also please see Holdridge 1947 for the classic determination of world plant formations - tropical dry forest is contrasted with tropical wet forest. 

at line 201 change to 'at 97% similarity,'

                Line 202: Text has been changed to ‘97% similarity’.

at line 213;; 'function of'

                Line 214: Text has been changed to ‘function of’.

at line 215; 'although'

                Line 216: Text has been changed to ‘although’.

at line 226; 'heterogeneous'

                Line 227: Text has been changed to ‘heterogenous’.

248; 'distribution'

Line 249: Text has been changed to ‘distribution’.

'pathway,' at line 255;

                Line 256: Text has been changed to ‘pathway’.

'Hevea' at line 305;

                Line 305: Text has been changed to ‘Hevea’.

'suggest' at line 343.

                Line 342: Text has been changed to ‘suggest’.

Moreover, I propose to change line 127 like this '.. at La Selva Biological Research Station [23], where two fully expanded leaves were selected ..', considering that author's name is part of ref. [23].

Line 127: Text has been changed to correct citation format given that the name is part of the reference.

For the same reason, delete 'Higgins et al., 2011' at line 139.

                Line 139: Higgins et al has been deleted from the text.

Ref. [15] is inappropriate for statement at line 49?

                Line 48: Proper citation has been included.

Finally, abbreviated journal names have to be used throughout the reference list.

                Lines 388-524: Journal names have been changed to abbreviated style.

  1. Unterseher, M.; Gazis, R.; Chaverri, P.; Guarniz, C.F.G.; Tenorio, D.H.Z. Endophytic fungi from Peruvian highland and lowland habitats form distinctive and host plant-specific assemblages. Biodiversity and conservation 2013, 22, 999-1016.
  2. Higgins, K.L.; Coley, P.D.; Kursar, T.A.; Arnold, A.E. Culturing and direct PCR suggest prevalent host generalism among diverse fungal endophytes of tropical forest grasses. Mycologia 2011, 103, 247-260.
  3. Del Olmo-Ruiz, M.; Arnold, A.E. Interannual variation and host affiliations of endophytic fungi associated with ferns at La Selva, Costa Rica. Mycologia 2014, 106, 8-21, doi:10.3852/13-098.
  4. Donald, J.; Roy, M.; Suescun, U.; Iribar, A.; Manzi, S.; Péllissier, L.; Gaucher, P.; Chave, J. A test of community assembly rules using foliar endophytes from a tropical forest canopy. Journal of Ecology 2020.
  5. Arnold, A.E.; Lutzoni, F. Diversity and host range of foliar fungal endophytes: are tropical leaves biodiversity hotspots? Ecology 2007, 88, 541-549, doi:10.1890/05-1459.
  6. Mooney, H.; Bullock, S.; Ehleringer, J. Carbon isotope ratios of plants of a tropical dry forest in Mexico. Functional Ecology 1989, 137-142.

Round 2

Reviewer 1 Report

The manuscript looks good in its present form.